# Operator Learning for High-Dimensional Symplastic Growth Dynamics with Stochastic Cell Division

**Ulyana Zubairova**
Department of Information Technologies,
Novosibirsk State University,
Novosibirsk, Russia
`u.zubairova@g.nsu.ru`

## Abstract

We study operator learning for a nonlinear dynamical system describing symplastic plant leaf growth with multiple interacting cell files and stochastic cell division. The biomechanical model consists of coupled ordinary differential equations governing visible cell lengths, relaxed wall lengths, isosmotic lengths, and shared wall fragments. For $N$ cell files with $M$ cells per file, the state dimension scales as $D(N, M) \sim 3NM + K(N, M)$, where $K$ denotes the number of shared fragments. While the number of state variables grows linearly in $N$, fragment-based mechanical coupling induces a rapidly increasing interaction structure, leading to dense Jacobians and growing computational cost of numerical integration. In the multi-file regime, repeated simulation becomes computationally prohibitive for parameter exploration and inverse calibration. We formalize the simulator as a nonlinear operator $\mathcal{F} : \Theta \subset \mathbb{R}^p \to \mathbb{R}^B$ mapping mechanical parameters to the longitudinal cell length profile. We train multilayer perceptron (MLP) surrogates to approximate $\mathcal{F}$ using simulator-generated data. The learned surrogate replaces repeated ODE integration and enables fast prediction of spatial growth profiles. We evaluate generalization performance on held-out parameter configurations and demonstrate efficient parameter calibration to experimental profiles. We further analyze structural properties of the parameter-to-profile map, including local regularity and an intrinsic stochastic noise floor induced by random cell division. Our results show that neural operator approximation provides a scalable framework for accelerating analysis and inverse modeling of coupled high-dimensional biological growth dynamics.

## 1 Introduction

Understanding how plant leaves grow under mechanical and osmotic constraints is central to developmental biology and crop modeling. Cell-based models of symplastic growth represent the tissue as a network of mechanically coupled cells sharing wall fragments. In particular, Zubairova et al. (2016) proposed a quasi-one-dimensional biomechanical model that explicitly expresses osmotic and turgor pressures and reproduces experimentally observed longitudinal cell length distributions in the wheat leaf epidermis.

From a dynamical systems perspective, this framework defines a high-dimensional hybrid stochastic ODE. Continuous biomechanical evolution of cell lengths and wall properties is coupled with discrete cell division events, while stochastic division factors introduce variability in the resulting profiles. For $N$ cell files with $M$ cells per file, the state dimension scales as $D(N, M) \sim 3NM + K(N, M)$, where $K(N, M)$ is the number of shared fragments. Fragment-mediated mechanical coupling produces dense Jacobians, and the computational cost of numerical integration increases with the number of coupled cell files. Repeated simulation therefore becomes impractical for parameter exploration and inverse calibration, a setting where machine learning surrogates have been successfully used to approximate mechanistic models in systems biology, including ODE-,

SDE-, and PDE-based models (Gherman et al., 2023). We contribute an operator-learning formulation for this high-dimensional symplastic growth setting.

To address this limitation, we adopt an operator-learning viewpoint. We formalize the simulator as a parameter-to-profile operator

$$\mathcal{F} : \Theta \subset \mathbb{R}^p \to \mathbb{R}^B,$$

which maps mechanical parameters to the longitudinal cell length profile obtained by spatial binning. Learning an approximation of this operator enables fast prediction without integrating the full high-dimensional state trajectory.

Beyond computational acceleration, this viewpoint clarifies structural properties of the model. The simulator defines a hybrid stochastic dynamical system whose parameter-to-profile map is locally Lipschitz away from changes in the division event schedule, while stochastic cell division induces intrinsic output variability. We show that this variability establishes an irreducible noise floor that lower-bounds the achievable surrogate error. In addition, we empirically quantify computational scaling with respect to the number of cell files and fit a power law $T(N) \approx aN^\gamma$ to simulation runtimes, obtaining $\gamma \approx 0.67$ with $R^2 \approx 0.93$. Although sublinear, this growth still leads to rapidly increasing cost in multi-file regimes and motivates replacing repeated ODE solves by a learned surrogate in calibration workflows.

**Contributions.** Our contributions are threefold. (i) We formalize symplastic leaf growth simulation as a parameter-to-profile operator of a high-dimensional hybrid stochastic system and analyze its local regularity. (ii) We establish an intrinsic stochastic error floor induced by random cell division. (iii) We empirically quantify computational scaling and demonstrate that neural surrogates provide substantial acceleration for inverse calibration while preserving profile accuracy.

## 2 MATHEMATICAL MODEL

We use the mechanical framework of Zubairova et al. (2016) for symplastic unidirectional growth of a linear leaf epidermis. The leaf is represented as a "brickwork" of cell files (rows of cells along the leaf). Each cell is characterized by *visible length* $l$, *relaxed length* $l_r$ (cell wall length in the unstressed state), and *isosmotic length* $l_i$ (related to cell biomass and osmotic content). Neighboring cells share wall fragments; the growth rate of a common fragment is determined by the sum of the free growth rates of the cells that contain it, so that cells influence each other mechanically (symplastic growth).

### 2.1 BIOMECHANICS OF A SINGLE CELL

Following Zubairova et al. (2016), the difference between osmotic pressure inside and outside the cell is expressed in terms of lengths as

$$P_{\text{osm}} = \alpha \, \frac{l_i - l}{l}, \tag{1}$$

where $\alpha$ is the coefficient of osmotic pressure (e.g., $\alpha = c_{\text{out}}RT$ from the Van't Hoff equation). Turgor pressure (hydrostatic pressure due to wall stress) is

$$P_{\text{turg}} = \beta \, \frac{l - l_r}{l_r}, \tag{2}$$

where $\beta = (S_w/S_c)E$ depends on the wall cross-section $S_w$, cell cross-section $S_c$, and Young's modulus $E$. Similarly to multicellular Lockhart–Ortega-type models that couple water flux and cell wall mechanics (Cheddadi et al., 2019), our formulation explicitly expresses turgor and osmotic pressure and couples growth through shared wall fragments. The yield threshold $P_c$ and irreversible wall growth $l_r$ belong to the cell wall extensibility tradition (Cosgrove, 1993). Water flows into the cell when the water potential $\Psi_w = (P_{\text{osm}} - P_{\text{osm}}^{\text{out}}) - P_{\text{turg}} > 0$. The growth rate of the visible cell length is proportional to this flow. In unidirectional growth, with hydraulic conductivity $L_w$ and cell width $r$,

$$\frac{dl}{dt} = r \cdot l \cdot L_w \cdot (P_{\text{osm}} - P_{\text{turg}}), \tag{3}$$

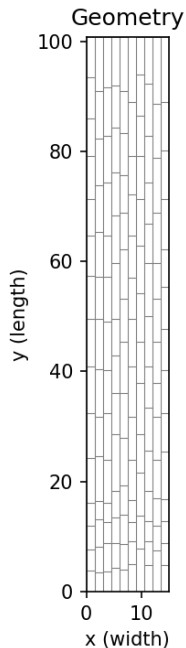
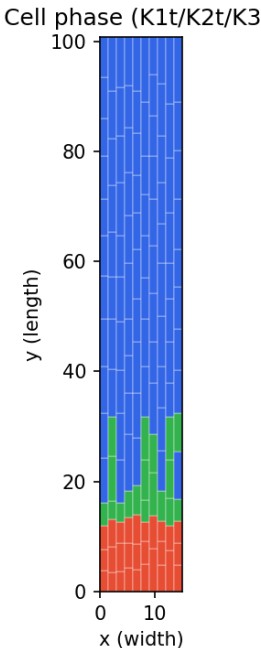

Figure 1: Multi-file leaf structure: cell files (rows along the leaf), shared transverse fragments (segments), and cell cycle phases along the longitudinal axis. Phase colors: K1t (red), K2t (green), K3t (blue).

or in specific form

$$\frac{1}{l}\frac{dl}{dt} = r \cdot L_w \cdot \left( \alpha \frac{l_i - l}{l} - \beta \frac{l - l_r}{l_r} \right). \tag{4}$$

Cell wall growth (irreversible change of relaxed length) is triggered when turgor exceeds a threshold $P_c$:

$$\frac{dl_r}{dt} = \begin{cases} 0, & \text{if } P_{\text{turg}} \leq P_c \\ \eta \frac{dl_i}{dt} (P_{\text{turg}} - P_c)^3, & \text{otherwise,} \end{cases} \tag{5}$$

where $\eta$ is the coefficient relating cell wall growth rate to biomass growth rate. The isosmotic length $l_i(t)$ is given by an explicit function of time (autonomous growth), e.g., piecewise-linear in the division zone (DZ) and elongation zone (EZ) with rates $a_1$, $a_2$ and critical value $l_{i,\max}$ (Zubairova et al., 2016).

## 2.2 Symplastic growth: fragments and cell files

The leaf epidermis is divided into transverse *fragments* (segments) indexed by $k$; fragment $k$ has length $\lambda_k$ and is shared by one cell from each file. The growth rate of the fragment is the average of the free specific growth rates of the cells that contain it:

$$\frac{d\lambda_k}{dt} = \frac{\lambda_k}{N} \sum_{n=1}^{N} \left( \frac{1}{l_{mn}} \frac{dl_{mn}}{dt} \right)_{\text{free}}, \quad (m : \lambda_k \in l_{mn}), \tag{6}$$

where $N$ is the number of cell files, $l_{mn}$ is the visible length of cell $m$ in file $n$, and the "free" rate is given by the right-hand side of equation 4 for that cell. The visible length of each cell is the sum of the lengths of the fragments belonging to it, so $dl_{mn}/dt$ is the sum of the corresponding $d\lambda_k/dt$. This couples all cells that share fragments and yields the symplastic dynamics. The multi-file geometry and cell phases along the leaf are illustrated in Figure 1.

## 2.3 Cell division in the division zone

Cells progress through three phases along the leaf: K1t (growth toward the division threshold), K2t (division zone, where $l_i$ may reach the critical value), and K3t (elongation, post-division). A cell in the division zone divides when its isosmotic length $l_i$ reaches a critical value $l_{i,\mathrm{max}}$. The mother cell is replaced by two daughter cells with length ratio $d/(1-d)$, where $d$ is the division factor (random, truncated normal in $(0.1, 0.9)$ with mean $0.5$). The initial isosmotic lengths of the daughters are $d \cdot l_{i,\mathrm{max}}$ and $(1-d) \cdot l_{i,\mathrm{max}}$, and their birth time $t_0$ is set to the division time (Zubairova et al., 2016). Segment (fragment) state is updated so that every segment reflects the new cell layout. This coupling between many cells and segments makes the multi-file regime both biologically relevant and numerically expensive. Because $d$ is stochastic, the same parameter vector can yield different $(L, t_f)$ across runs; in practice one may average over several runs with different random seeds for each $x$, or include the seed as an extra input to the surrogate. The present work uses a Python reimplementation of this framework (with the same equations and division rule) to generate training data for the operator surrogate.

## 2.4 State dimension and computational cost: the role of the number of cell files

A central structural parameter of the model is the *number of cell files* $N$: the leaf epidermis is represented as $N$ parallel rows of cells along the longitudinal axis, and each transverse fragment (segment) is shared by exactly one cell from each file, so $N$ cells are mechanically coupled at each fragment. For $N$ cell files and $M$ cells per file (at a given time), the state comprises $3NM$ cell-level variables (visible length $l$, relaxed length $l_r$, and isosmotic length $l_i$ per cell) plus $K(N, M)$ fragment lengths $\lambda_k$, where $K$ grows with the number of cells and their connectivity along the leaf. Thus $D(N, M) \sim 3NM + K(N, M)$. The dependence on $N$ is twofold: (i) the number of cell variables scales as $NM$, and (ii) each fragment couples $N$ cells, so the interaction structure and the number of nonzero entries in the Jacobian grow with $N$. As a result, numerical integration cost increases rapidly with the number of cell files. We explicitly quantify this dependence in the experiments (Sec. 4) by measuring state dimension and wall-clock time for several $(N, M)$ configurations. This motivates learning a map from parameters to observables (e.g., the cell length profile) without integrating the full state trajectory.

## 2.5 Parameter-to-profile map of the hybrid stochastic model

Let $z(t) \in \mathbb{R}^{D(N,M)}$ denote the full simulator state, consisting of cell-level variables $(l, l_r, l_i)$ and fragment lengths $\lambda$. For a parameter vector $x \in \Theta \subset \mathbb{R}^p$ (e.g., $x = (\alpha, \eta, P_c)$), the simulator defines a hybrid stochastic dynamical system with continuous flows and discrete state resets:

$$\dot{z}(t) = G(z(t); x), \qquad z(0) = z_0(x), \tag{7}$$

where discrete events correspond to cell division in the division zone. Division events are triggered when the isosmotic length of a cell reaches a threshold; after division, the state is updated via a reset map that depends on a random division factor $d$.

Let $\omega$ denote a realization of the simulator randomness (e.g., a random seed determining division factors). For each $(x, \omega)$, the simulator produces a terminal state $z(t_f(x, \omega))$, where $t_f(x, \omega)$ is the stopping time at which the total leaf length first reaches the prescribed target $L_y$.

We define the profile projection operator

$$\mathcal{P} : \mathbb{R}^{D(N,M)} \to \mathbb{R}^B,$$

which partitions the longitudinal axis into $B$ spatial bins and returns the corresponding mean cell lengths. The simulator thus induces a stochastic parameter-to-profile map

$$F_\omega : \Theta \to \mathbb{R}^B, \qquad F_\omega(x) = \mathcal{P}\big(z(t_f(x, \omega); x, \omega)\big). \tag{8}$$

In practice, experimental measurements correspond to averaged cell length profiles. We therefore consider the mean profile operator

$$\bar{F}(x) = \mathbb{E}_\omega \left[ F_\omega(x) \right], \tag{9}$$

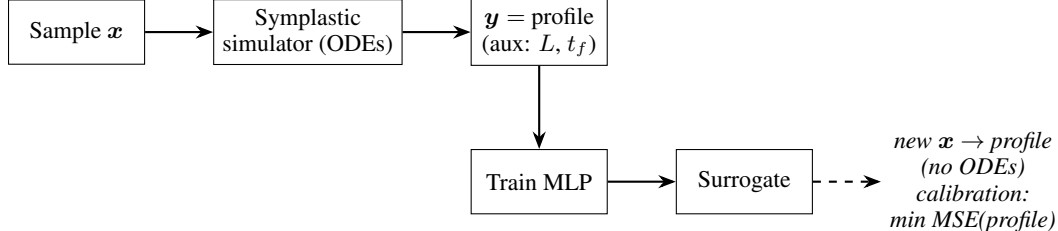

Figure 2: Operator learning pipeline. Top: sample $\boldsymbol{x} \in \Theta$, run the symplastic simulator; each run yields the profile (and auxiliary scalars $L$, $t_f$). The operator $\mathcal{F}$ maps parameters to the profile only. Bottom: train an MLP to approximate $\mathcal{F}$, then use $\widehat{\mathcal{F}}$ for fast prediction and calibration (minimize MSE between predicted and experimental profile) without solving the ODEs.

which maps mechanical parameters to the expected longitudinal profile.

This formulation makes explicit that surrogate learning therefore amounts to approximating the parameter-to-profile operator of a high-dimensional hybrid stochastic dynamical system.

## 3 OPERATOR LEARNING AND SURROGATE

We approximate the parameter-to-profile operator $\mathcal{F} : \Theta \subset \mathbb{R}^p \to \mathbb{R}^B$ mapping parameters to the longitudinal profile only; $\Theta$ is the set of admissible mechanical parameters (e.g., $\alpha$, $\eta$, $P_c$). The simulator also yields auxiliary scalar outputs $L$ (final leaf length) and $t_f$ (stopping time), which we may predict alongside the profile but are not part of the operator $\mathcal{F}$. The goal is to learn an approximation $\widehat{\mathcal{F}}$ to $\mathcal{F}$ from data $\{(\boldsymbol{x}^{(i)}, \boldsymbol{y}^{(i)})\}$ with $\boldsymbol{y}^{(i)} = \mathcal{F}(\boldsymbol{x}^{(i)})$, so that we can evaluate $\widehat{\mathcal{F}}(\boldsymbol{x})$ without integrating the ODEs.

### 3.1 PIPELINE

The pipeline is summarized in Figure 2. Parameter vectors $\boldsymbol{x} \in \Theta$ are sampled and fed to the symplastic simulator; each run yields the profile $(\bar{\ell}_1, \ldots, \bar{\ell}_B) \in \mathbb{R}^B$ and auxiliary scalar outputs $(L, t_f)$. The surrogate approximates $\mathcal{F}$ mapping $\boldsymbol{x} \to$ profile; the MLP may also predict $L$ and $t_f$ as auxiliary targets. The trained surrogate $\widehat{\mathcal{F}}$ then predicts the profile for new $\boldsymbol{x}$ without solving the ODEs. The main application is calibration: given an experimental profile binned to $B$ bins, we minimize the MSE between $\widehat{\mathcal{F}}(\boldsymbol{x})$ and the experimental profile over $\boldsymbol{x} \in \Theta$.

### 3.2 INPUTS AND OUTPUTS

We vary a small set of parameters; the rest (geometry, etc.) are fixed. The input vector $\boldsymbol{x}$ typically includes the law-of-growth parameters:

- $\alpha$ (coefficient of osmotic pressure, Eq. 1),
- $\eta$ (coefficient of cell wall growth rate, Eq. 5),
- $P_c$ (turgor threshold when wall growth begins, Eq. 5).

The set of calibratable parameters can be extended to include parameters of the growth function $l_i(t)$ (e.g., cell cycle time, elongation phase duration, division and elongation thresholds, min/max cell sizes). These are among the parameters with significant sensitivity (Zubairova et al., 2016). Each run of the simulator returns a final total leaf length $L$, the time $t_f$ at which the target length $L_y$ was reached, and a *cell length profile*: the leaf is divided into $B$ bins along the longitudinal axis (distance from the base), and the mean cell length in each bin is recorded. The operator $\mathcal{F}$ maps to the profile $(\bar{\ell}_1, \ldots, \bar{\ell}_B) \in \mathbb{R}^B$; $L$ and $t_f$ are auxiliary simulator outputs. Experimental data (cell lengths vs. distance from the leaf base) can be binned into the same $B$ bins to obtain a target profile $\boldsymbol{y}_{\exp}^{\mathrm{prof}}$. Calibration amounts to finding parameters $\boldsymbol{x}$ that minimize the MSE between the surrogate-predicted profile and $\boldsymbol{y}_{\exp}^{\mathrm{prof}}$, without running the full ODE solver in the inner loop.

## 3.3 Training data

We draw $N$ parameter vectors $\boldsymbol{x}^{(i)}$ from uniform distributions over prescribed bounds, run the simulator for each, and collect successful pairs $(\boldsymbol{x}^{(i)}, \boldsymbol{y}^{(i)})$. Failed runs (numerical errors or non-convergence) are discarded. The dataset is $\mathcal{D} = \{(\boldsymbol{x}^{(i)}, \boldsymbol{y}^{(i)})\}_{i=1}^{n}$. We split $\mathcal{D}$ into training (e.g. 80%) and test (20%) sets; surrogates are trained only on the training set and evaluated on the test set to report MAE and RMSE without leakage.

## 3.4 MLP operator approximation

A multi-layer perceptron (MLP) regressor is trained on $\mathcal{D}$ to approximate the operator $\mathcal{F}$: it maps $\boldsymbol{x} \in \Theta$ to the predicted profile $\widehat{\mathcal{F}}(\boldsymbol{x}) \in \mathbb{R}^{B}$ (and may optionally predict $L$, $t_f$ as auxiliary outputs). We use scikit-learn's `MLPRegressor` with two hidden layers (e.g., 32 units each) and early stopping. The MLP is fast at test time, which is essential for calibration: the optimizer (e.g., L-BFGS-B) repeatedly evaluates the loss (MSE between predicted and experimental profile) and thus calls $\widehat{\mathcal{F}}$ many times without running the ODE solver.

**Local regularity.** **Proposition 1 (Piecewise local Lipschitz continuity).** Assume that for a fixed $(x_0, \omega)$ the event schedule (the number and order of cell divisions) does not change in a neighborhood $U \subset \Theta$ of $x_0$, and that the vector field $G(\cdot; x)$ is locally Lipschitz in $z$ and continuous in $x$. Assume further that the reset maps at division events are locally Lipschitz in the pre-event state.

Then there exists a constant $L_{x_0, \omega}$ such that for all $x_1, x_2 \in U$,

$$\|F_\omega(x_1) - F_\omega(x_2)\| \leq L_{x_0, \omega} \|x_1 - x_2\|.$$

*Sketch of argument.* Stability of ODE flows under Lipschitz vector fields implies continuous dependence of trajectories on parameters. Since reset maps are Lipschitz and the event schedule is fixed, the full hybrid flow remains locally Lipschitz. The projection $\mathcal{P}$ preserves Lipschitz continuity.

*Remark.* Changes in the event schedule may occur near parameter values where division thresholds are reached at different times. Such regions introduce non-smoothness in the parameter-to-profile map and increase approximation difficulty.

**Intrinsic stochastic error floor.** **Proposition 2 (Irreducible error due to stochastic division).** Let $Y = F_\omega(x) \in \mathbb{R}^{B}$ be the stochastic profile at fixed $x$, with mean $\mu(x) = \mathbb{E}[Y]$ and covariance matrix $\Sigma(x)$.

For any deterministic predictor $g : \Theta \to \mathbb{R}^{B}$,

$$\mathbb{E}_\omega \|g(x) - Y\|_2^2 = \|g(x) - \mu(x)\|_2^2 + \operatorname{tr} \Sigma(x) \geq \operatorname{tr} \Sigma(x).$$

Therefore, the trace of the output covariance acts as an intrinsic noise floor: no deterministic surrogate can achieve mean squared error below the variance induced by stochastic cell division.

**Implications for surrogate learning.** Proposition 1 suggests that away from event-schedule transitions, the parameter-to-profile map is locally regular, which justifies approximation by smooth function classes such as MLPs. Proposition 2 shows that even a perfect surrogate cannot achieve error below the intrinsic stochastic variance of the simulator. Together, these results clarify both the feasibility and the fundamental limits of neural operator approximation in this hybrid stochastic setting.

## 4 Experiments

We approximate the mean parameter-to-profile operator $\bar{\mathcal{F}} : \Theta \to \mathbb{R}^{B}$ using a neural surrogate trained on simulator-generated samples.

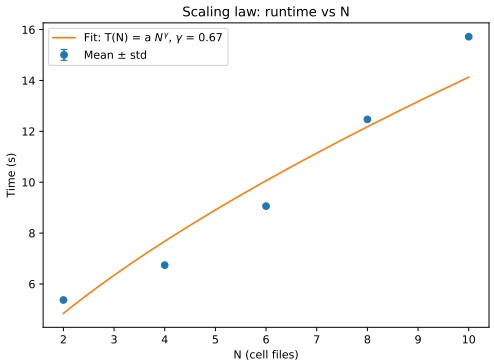
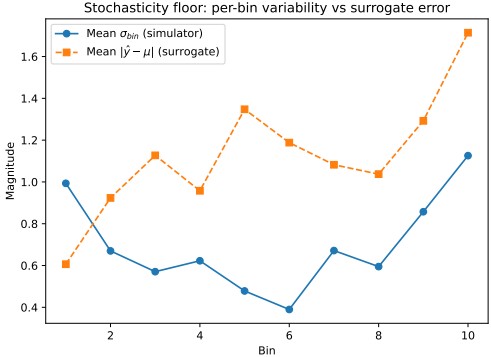

(a) Scaling: runtime vs $N$ and power-law fit ($\gamma \approx 0.67$, $R^2 \approx 0.93$).

(b) Stochasticity floor: surrogate RMSE vs. within-simulator $\sigma$.

Figure 3: (a) Simulation runtime vs number of cell files $N$. (b) Surrogate profile RMSE vs. within-simulator variability for fixed parameter vectors (ratio $\leq 1$: surrogate within stochastic spread).

## 4.1 SETUP

We use the multi-file configuration: $N = 10$ cell files, $M = 4$ cells per file initially, cell division enabled, target length $L_y = 100$ (in model units). Each simulation returns the profile ($B = 10$ spatial bins) and auxiliary scalars $L$ and $t_f$. Parameter ranges: $\alpha \in [5, 15]$, $\eta \in [0.05, 0.3]$, $P_c \in [1, 4]$ bar (Zubairova et al., 2016). The dataset consists of $n = 200$ successful runs, split 80/20 into train and test. For each parameter vector we use a single simulator run (one random seed). We train an MLP and, for comparison, a Ridge regression baseline on the same inputs and outputs.

## 4.2 SIMULATOR SCALING: DEPENDENCE ON THE NUMBER OF CELL FILES

Because the number of cell files $N$ is a key structural parameter (state dimension and fragment coupling both scale with $N$), we explicitly measure how state dimension $D$ and wall-clock time per simulation depend on $N$ (and on $M$). We run the simulator for several configurations $(N, M)$ (e.g., $(1, 4)$, $(2, 4)$, and optionally $(2, 8)$ or $(4, 4)$) and record the state dimension (number of cell-level variables plus fragment lengths) and mean runtime. State dimension $D$ and mean runtime for each configuration are reported in Table 1; we discuss the observed scaling with $N$ (and with $D$). This dependence justifies the use of operator learning: when $N$ or $M$ grows, repeated simulation becomes prohibitive for calibration and parameter studies.

We further summarize the empirical scaling by fitting a power law $T(N) \approx aN^\gamma$ to the runtimes in Table 1 via log–log regression (using the reported times for $N = 2, 4, 6, 8, 10$). This yields $\gamma \approx 0.67$ with $R^2 \approx 0.93$, indicating a sublinear but steadily increasing computational cost with the number of coupled cell files (Figure 3a). This motivates replacing repeated ODE solves by the learned surrogate in parameter studies and calibration.

## 4.3 PROFILE PREDICTION QUALITY

We evaluate the surrogate on the held-out test set. On the test set we obtain MAE (RMSE) of 20.29 (24.23) for $L$, 5.76 (6.73) for $t_f$, and 1.23 (1.50) for the binned profile. We also report $R^2$ for the profile and maximum absolute error across bins. We plot 5–10 example profiles (true vs. predicted) to assess shape fidelity. Permutation feature importance for $\alpha$, $\eta$, $P_c$ on the profile indicates that $\alpha$ has the strongest influence, followed by $P_c$ and $\eta$. If a baseline is trained, we include its metrics in the same table.

## 4.4 STOCHASTICITY OF THE SIMULATOR

Because cell division is stochastic, the same parameter vector $x$ yields different profiles across runs. For five fixed parameter vectors $x$ we run the simulator with $S = 10$ different seeds and

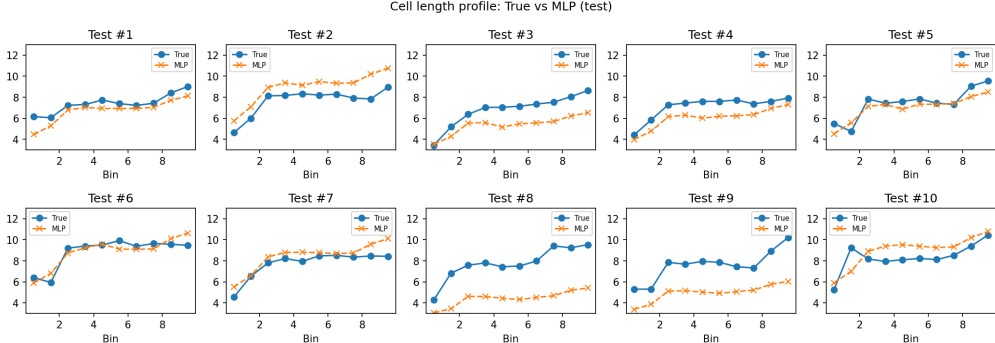

Figure 4: Example cell length profiles: true (from simulator) vs. MLP-predicted on the test set. Each subplot is one test configuration.

compute the per-bin standard deviation of the profile. We compare the surrogate's test RMSE to this within-simulator variability: when the surrogate error is comparable to or smaller than the typical run-to-run spread, the approximation is adequate for calibration. This empirical variability provides an estimate of the intrinsic stochastic noise floor described in Proposition 2. Figure 3b illustrates this comparison for our setup.

## 4.5 CALIBRATION: SYNTHETIC INVERSE TEST

We perform a synthetic inverse test: choose a "true" parameter vector $x^*$, generate a target profile $y^*$ (one run or mean over a few runs), add small Gaussian noise to mimic measurement error, and run calibration by minimizing MSE between the surrogate-predicted profile and the noisy target over $x$. We report the parameter error $|\hat{x} - x^*|$ (per component or norm), the profile MSE at the recovered $\hat{x}$, the number of loss evaluations, and wall-clock time. We compare with the cost of calibration using the full simulator when feasible (e.g., few iterations) and report speedup. Figure 5a shows an example: target, noisy target, and profile after calibration.

This demonstrates that surrogate-based calibration can recover parameters within the intrinsic stochastic variability of the model.

## 4.6 COMPUTATIONAL EFFICIENCY

We report: (i) mean wall-clock time for one full simulation; (ii) time for $10^4$ surrogate evaluations; (iii) time for one calibration run (surrogate-only). In our setup, one simulator run for $(N, M) = (2, 4)$ takes on the order of 4 s, while $10^4$ surrogate evaluations take on the order of 5 ms, yielding a speedup of $\sim 10^6$ per evaluation in our setup; one calibration run takes on the order of 15 ms. The speedup factor and the cost of calibration with the surrogate summarize the practical benefit of the learned operator.

## 4.7 RESULTS SUMMARY

State dimension $D$ and mean runtime per simulation for configurations $(N, M)$ from $(2, 4)$ to $(10, 4)$ are reported in Table 1; the fitted power law yields $\gamma \approx 0.67$ with $R^2 \approx 0.93$ (see Figure 3a), confirming that computational cost grows with the number of cell files and motivating the use of a surrogate for parameter studies. Test-set metrics for the MLP are MAE (RMSE) 20.29 (24.23) for $L$, 5.76 (6.73) for $t_f$, and 1.23 (1.50) for the profile. Example true vs. predicted profiles are shown in Figure 4. The stochasticity floor is illustrated in Figure 3b; synthetic inverse calibration and the learning curve are shown in Figure 5a and Figure 5b, supporting the use of larger datasets or active learning in future work.

Profile prediction accuracy is sufficient for calibration: the surrogate's profile RMSE is comparable to or below the within-simulator variability (stochasticity floor) when the same parameters are run with different division seeds, consistent with Proposition 2. The synthetic inverse test shows that

Table 1: Dependence on the number of cell files $N$ and cells per file $M$: state dimension $D$ and mean wall-clock time per simulation. $N$ is the number of cell files (parallel rows); each fragment couples $N$ cells.

| $(N, M)$ | State dim. $D$ | Time (s) |
|----------|----------------|----------|
| $(2, 4)$ | 55 | 5.37 |
| $(4, 4)$ | 106 | 6.74 |
| $(6, 4)$ | 160 | 9.06 |
| $(8, 4)$ | 211 | 12.47 |
| $(10, 4)$ | 264 | 15.72 |

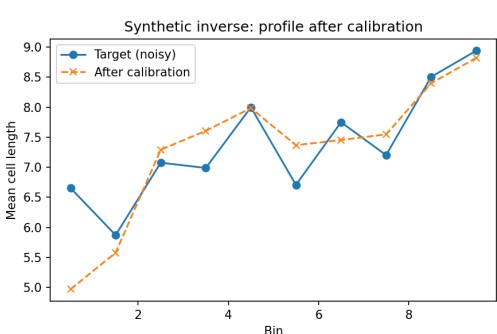

(a) Synthetic inverse: target, noisy target, and profile at $\hat{\boldsymbol{x}}$.

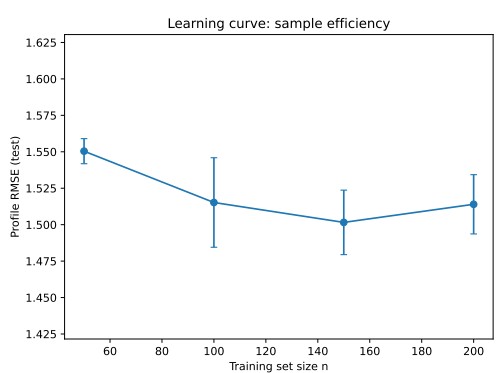

(b) Profile RMSE vs. training set size.

Figure 5: (a) Synthetic inverse test: profile after calibration. (b) Learning curve: profile RMSE decreases with training set size, supporting active learning.

parameters can be recovered by minimizing MSE between the surrogate-predicted profile and a noisy target, with the recovered profile matching the target within the intrinsic stochastic spread. The learned surrogate is orders of magnitude faster than one ODE solve per evaluation (speedup $\sim 10^6$ per evaluation in our setup), so calibration over $\boldsymbol{x} \in \Theta$ is feasible in practice without running the full simulator in the inner loop.

## 5 CONCLUSION

We studied operator learning for a high-dimensional symplastic leaf growth model with stochastic cell division (Zubairova et al., 2016). The state dimension scales as $D(N, M) \sim 3NM + K(N, M)$, and fragment-based mechanical coupling leads to dense interactions, making repeated numerical simulation increasingly costly as the number of cell files grows.

In line with surrogate-modelling practice for biological systems (Gherman et al., 2023), we justified the surrogate approach by explicitly quantifying computational scaling with respect to the number of cell files and by accounting for the intrinsic stochasticity floor induced by random cell division. We formalized the simulator as a nonlinear operator $\mathcal{F} : \Theta \to \mathbb{R}^B$ mapping mechanical parameters to the longitudinal cell length profile, and trained an MLP surrogate to approximate $\mathcal{F}$.

The learned surrogate replaces repeated ODE integration with fast forward evaluation and enables efficient calibration to experimental profiles by minimizing MSE over the parameter space. We reported generalization performance on held-out configurations and analyzed feature importance to assess parameter influence on the predicted profiles. Overall, our results indicate that neural operator approximation provides a computationally scalable framework for accelerating analysis and inverse modeling of coupled high-dimensional biological growth dynamics.

Consistent with recommendations from surrogate-modelling practice (Gherman et al., 2023), natural next steps include active learning to add training points in regions of higher surrogate error, dimensionality reduction of inputs or outputs to improve data efficiency, and more explicit treatment of stochasticity (e.g., averaging over division factors or incorporating the random seed as an input). Further directions include optimization over the surrogate (e.g., maximizing $L$ for fixed time (Salehi et al., 2020)) and validation on real experimental profiles.

Code and reproduction instructions are available upon request.

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
