# OpenReview forum: "Operator Learning for High-Dimensional Symplastic Growth Dynamics with Stochastic Cell Division"
_mathai.club/MathAI/2026/Conference — 2026 Oral_

### Official Review · Reviewer_8Msh · 2026-03-11
**review of 'Operator Learning for High-Dimensional Symplastic Growth Dynamics with Stochastic Cell Division.'**

**Rating:** 6
**Confidence:** 5

**Review:**

This paper studies operator learning for a high dimensional nonlinear dynamical system describing symplastic plant leaf growth with multiple interacting cell files and stochastic cell division. The authors formalize an existing biomechanical ODE model (Zubairova et al., 2016) as a parameter to profile operator $\mathcal{F}: \Theta \subset \mathbb{R}^p \rightarrow \mathbb{R}^B$ mapping mechanical parameters ($\alpha, \eta, P_c$) to the longitudinal cell length profile. They analyze the model's structural properties, proving local Lipschitz continuity of the operator away from event schedule changes (Proposition 1) and establishing an intrinsic stochastic error floor due to random cell division: $\mathbb{E}_\omega|\mathcal{g}(x)-Y|^2 \geq \operatorname{tr}\Sigma(x)$ (Proposition 2). To address the computational cost of repeated ODE integration, which scales with the number of cell files $N$ as $T(N) \propto N^{0.67}$ (Figure 3a), they train an MLP surrogate to approximate $\mathcal{F}$, enabling fast parameter calibration with speedups of several orders of magnitude.

The paper has several notable strengths. First, it makes genuine mathematical contributions aligned with the conference mission. Proposition 1 formally justifies approximation by smooth functions, while Proposition 2 provides a fundamental lower bound on surrogate accuracy directly linked to the model's inherent stochasticity. Second, the problem formulation is exceptionally rigorous. Section 2 clearly presents the coupled ODEs (Equations 1 through 6), defines the state dimension scaling $D(N,M) \sim 3NM + K(N,M)$, and formalizes the stochastic operator $\mathcal{F}_\omega(x)$. Third, the work is well motivated by quantitative scaling analysis (Table 1, Figure 3a) and demonstrates principled handling of stochasticity by explicitly modeling randomness, proving its theoretical consequence, and empirically validating that surrogate error matches the intrinsic noise floor (Figure 3b). The experimental validation is thorough, including profile prediction metrics (RMSE 1.50), learning curves, synthetic inverse calibration (Figure 5a), and clear speedup quantification.

However, the paper has some limitations. The proofs of Propositions 1 and 2 are presented as sketches rather than fully rigorous treatments. For instance, the argument for local Lipschitz continuity would benefit from explicit constants or references to hybrid system stability theorems. Additionally, the study uses only a standard small MLP (two layers of 32 units) and does not compare with more specialized neural operator architectures such as DeepONet or Fourier Neural Operator that might better exploit the problem structure. Finally, the dataset of 200 simulations is relatively modest. These limitations do not outweigh the paper's contributions but explain the rating.
Another important weakness is the relatively shallow literature review. The paper cites only a limited set of works and does not sufficiently engage with recent developments in neural operator learning and surrogate modeling. As a result, it is difficult to clearly assess the novelty of the contribution relative to the current state of the field.
Due to the aforementioned reasons, I feel that this paper does have a chance of being accepted.

---

### Official Review · Reviewer_DMCe · 2026-03-13
**Operator Learning for High-Dimensional Symplastic Growth Dynamics with Stochastic Cell Division**

**Rating:** 7
**Confidence:** 4

**Review:**

This article represents a high-quality study, fully consistent with the conference's scope and high standards. It is distinguished by a clear mathematical formalization of a complex hybrid stochastic system as an operator mapping mechanical parameters to a cell length profile. The authors go beyond the applied application of neural networks and conduct a theoretical analysis, proving the local Lipschitz property of the parametric mapping and establishing a fundamental limit on the accuracy of any deterministic surrogate, conditioned by the inherent stochasticity of the cell division model. These theoretical results are supported by carefully designed experiments: the scaling of computational complexity depending on the number of cell strands is quantitatively assessed, the error of the trained MLP surrogate is shown to lie within the natural stochastic variance of the model itself, and the use of the surrogate for inverse calibration demonstrates impressive speedup—hundreds of thousands of times compared to direct multiple integration of the original system. This work makes a significant contribution to both computational biology and machine learning methodology, offering an effective tool for analyzing complex growth processes.
However, several aspects could be improved. First, the architecture used—a simple multilayer perceptron—while sufficient for this task due to the small input dimensionality, is inferior to modern operator architectures (e.g., DeepONet or the Fourier Neural Operator) in potential flexibility and sampling invariance. The authors should either justify their choice of MLP in more detail or discuss the prospects for applying more advanced methods. Second, the surrogate was trained using single implementations of the stochastic model, while the target operator is defined as the expected value over random divisions. This can lead to bias when calibrating to experimental data, which typically represent population averages. Although this does not manifest itself in synthetic tests, for real-world applications, it would be worthwhile to either average several runs when generating the training set or use a probabilistic surrogate.
These remarks do not detract from the overall high rating of the work, but rather point to areas for further development. The article will undoubtedly be of interest to a wide audience at the conference and deserves acceptance.

---

### Decision · Program_Chairs · 2026-03-14

**Decision:**

Accept (Oral)

**Comment:**

Dear Author(s),

On behalf of the Program Committee of the International Conference on Mathematics of Artificial Intelligence (MathAI 2026), we are pleased to inform you that your paper has been accepted for an oral presentation at MathAI 2026.

Your paper was evaluated through a rigorous two-stage review process involving both automated screening and expert review by members of the Program Committee. The reviewers recognized the quality and contribution of your work.

Presentation details:

- Format: Oral presentation (15–20 minutes + 5 minutes Q&A)
- Mode: You may present either in person (offline) at the conference venue in Sirius, Russia, or remotely via Zoom. Please indicate your preferred mode when confirming your participation.
- Conference dates: Marh 30 - April 3, 2026
- Website: https://mathai.club

Next steps:

1. Please confirm your participation and presentation mode by replying to this email mathai.club@yandex.ru no later than March 15, 2026 18:00 Moscow time.
2. If you plan to attend in person, the organizing committee will provide accommodation details separately.
3. Please prepare your final camera-ready manuscript according to the formatting guidelines available at https://mathai.club and upload it to OpenReview by March 15, 2026 18:00 Moscow time.

Should you have any questions regarding the program, logistics, or your presentation slot, please do not hesitate to contact us.

We look forward to your contribution to MathAI 2026.

With kind regards,

MathAI 2026 Program Committee
International Conference on Mathematics of Artificial Intelligence
https://mathai.club
OpenReview: https://openreview.net/group?id=mathai.club/MathAI/2026/Conference
Telegram: https://t.me/MathAI_club
Email: mathai.club@yandex.ru